# Aging of the Immune System: Focus on Natural Killer Cells Phenotype and Functions

**DOI:** 10.3390/cells11061017

**Published:** 2022-03-17

**Authors:** Ashley Brauning, Michael Rae, Gina Zhu, Elena Fulton, Tesfahun Dessale Admasu, Alexandra Stolzing, Amit Sharma

**Affiliations:** 1SENS Research Foundation, Mountain View, CA 94041, USA; ashley.brauning@sens.org (A.B.); michael.rae@sens.org (M.R.); ginazhu2@gmail.com (G.Z.); elenafulton@gmail.com (E.F.); tesfahun.admasu@sens.org (T.D.A.); 2Centre for Biological Engineering, Wolfson School of Electrical, Material and Manufacturing Engineering, Loughborough University, Loughborough LE11 3TU, UK

**Keywords:** natural killer cells (NK cells), aging, immunosenescence, senescence, elderly, frailty, cytokines, inflammation, immune system

## Abstract

Aging is the greatest risk factor for nearly all major chronic diseases, including cardiovascular diseases, cancer, Alzheimer’s and other neurodegenerative diseases of aging. Age-related impairment of immune function (immunosenescence) is one important cause of age-related morbidity and mortality, which may extend beyond its role in infectious disease. One aspect of immunosenescence that has received less attention is age-related natural killer (NK) cell dysfunction, characterized by reduced cytokine secretion and decreased target cell cytotoxicity, accompanied by and despite an increase in NK cell numbers with age. Moreover, recent studies have revealed that NK cells are the central actors in the immunosurveillance of senescent cells, whose age-related accumulation is itself a probable contributor to the chronic sterile low-grade inflammation developed with aging (“inflammaging”). NK cell dysfunction is therefore implicated in the increasing burden of infection, malignancy, inflammatory disorders, and senescent cells with age. This review will focus on recent advances and open questions in understanding the interplay between systemic inflammation, senescence burden, and NK cell dysfunction in the context of aging. Understanding the factors driving and enforcing NK cell aging may potentially lead to therapies countering age-related diseases and underlying drivers of the biological aging process itself.

## 1. Introduction

Aging is the most significant cause of morbidity and mortality in the developed world, and increasingly in developing countries as well [1,2]. While the decline in smoking prevalence and advancements in medical science have resulted in an increased lifespan, there is an increasing population prevalence of the diseases and debilities of aging associated with the not-as-rapid increase in health span [2,3]. Immunosenescence, the age-related decline in immune function, is a critical cause of age-related ill-health, driving dramatic increases in vulnerability to infectious disease with age (including the dramatic age-related rise in morbidity and mortality from COVID-19) [4,5]. In addition to immunosenescence, chronic sterile low-grade inflammation associated with aging (“inflammaging”) is likely contributing to the pathogenesis of multiple chronic diseases of aging [6,7].

NK cells are large granular innate immune cells that belong to the family of group 1 innate lymphocytes (ILC1) [8]. They play a vital role as immunological first responders, rapidly eliminating cells infected with viruses or other intracellular pathogens as well as pre-malignant cells [9], and have recently been identified as the principal actors in immune surveillance of senescent cells [10,11,12], which accumulate with age and are implicated in inflammaging, several age-related diseases of aging, and the underlying biological aging process itself [13,14]. Yet, comparatively little is known about the effects of aging on NK cells as compared to the cells in the adaptive immune system [15]. In this review, we discuss what is known about the phenotype and drivers of age-related NK cell dysfunction, areas of uncertainty, potential sources of new insight, and approaches to NK cell immune rejuvenation and fortification. Understanding and remediating the age-related decline in NK cell function could be an important means of ameliorating critical proximate causes of age-related ill-health and death and opposing one of the underlying drivers of aging.

## 2. NK Cells in Immunity: An Overview

NK cells eliminate target cells through direct cell-to-cell contact-based NK cell cytotoxicity (NKCC) and by secreting cytokines, such as interferon gamma (INF-γ) or tumor necrosis factor alpha (TNF-α) [16,17], which recruit additional immune cells to the site of infection or inflammation. NK cells represent ≈5–10% of peripheral blood lymphocytes (PBMC), with approximately 2 billion NK cells circulating in adult humans [18]. NK cells reside in different tissues, including bone marrow, lymph nodes, liver, skin, lung, and, to a lesser extent, secondary lymphoid organs [19]; however, this has not been extensively studied.

The human NK cell population can be subdivided based on the expression of the surface markers CD56 and CD16, with CD56^dim^ NK cells being largely cytotoxic and CD56^bright^ NK cells primarily responsible for the secretion of cytokines, such as IFN-γ and TNF-α [20,21]. However, in absolute numbers, CD56^bright^ NK cells, located predominantly in peripheral and lymphoid tissues [22,23], outnumber CD56^dim^ NK cells. The CD56^dim^ NK cell subpopulation represent at least 90% of peripheral blood NK cells, whereas the CD56^bright^ NK cells constitute the remaining [24]. NK cell phenotypes and their changes with age are described in greater depth in the subsequent sections.

The process of NK cell differentiation and maturation has been extensively reviewed elsewhere and is beyond the scope of this review [25]. In a brief summary, human NK cells develop from common lymphoid progenitor (CLP) cells derived from CD34^+^ hematopoietic stem cells (HSCs), the common precursors of NK, T, B, and other lymphoid cells [26,27]. CLP cells express different markers, including CD38, CD7, CD10, CD127, and CD122. The expression of CD122 (IL-2Rβ/IL-15R) marks the irreversible fate of CLPs starting to differentiate into NK lineage [28,29,30]. NK cells can act as a bridge between innate and adaptive immunity, as they can interact with T and B cells via their costimulatory ligands, such as CD40L and OX40L, which further promote NK cell differentiation [31,32].

NK cells interact with target cells via activating and inhibitory receptors, which bind to the corresponding ligands on the target cell’s surface. Depending on the balance of these ligand–receptor interactions, the NK cell determines whether or not to eliminate the target cell [33]. This review focuses on the alteration of the function of human NK cells with age; the NK cell receptors in humans and rodents have been extensively reviewed elsewhere [9,34]; we will therefore only provide a brief summary of this topic, as a background for what follows.

NKG2D, Ly49, and natural cytotoxicity receptors (NCRs), such as NKp46, are some of the main activating receptors for NK cells [16,35]. When the NKG2D or Ly49 interact with target cell ligands, such as MICA/MICB or ULBPs [36], their signal is mediated intracellularly through the YINM and immunoreceptor tyrosine-based activation motif (ITAM), respectively [36,37,38]. The phosphorylation of YINM or ITAM by tyrosine kinases leads to a cascade that results in the release of perforins and granzymes from NK cells to induce apoptosis in target cells [37,38]. In addition, NK cells also secrete cytokines and chemokines, which mediate their effector function directly on the target cells (cytotoxic signal) or through the recruitment of other immune cells, such as macrophages and neutrophils, to the site of inflammation [39,40].

The killer cell immunoglobulin-like receptors (KIR) and NKG2A are the major families of inhibitory receptors expressed on NK cells [41]. KIR bind to the HLA-A/B/C ligand, whereas NKG2A recognizes the inhibitory HLA-E [42,43]. These ligands bind to their respective receptors via immunoreceptor tyrosine-based inhibitory motifs (ITIM), leading to the inactivation of NK cells [44]. In addition, KIR and NK2GA are also known to coordinate NK cell tolerance [45]. Major histocompatibility complex class I (MHC-I) present non-self antigens for target engagement by CD8+ T-cells, but, conversely, are essential ligands for inhibitory receptors on NK cells for both mice and humans [46]. NK cells can recognize MHC- I directly (class Ia) or indirectly (class Ib). Reduced expression of MHC-I in cancer cells or cells infected with certain viruses lowers the inhibitory bar for attack by NK cells, enabling these cells’ elimination [47].

Mice and humans differ in the mechanism by which MHC class Ia is recognized: human NK cells use KIR to recognize MHC Ia, while mice use lectin-like receptors of the Ly49 family [48,49,50]. However, mice and humans have similar mechanisms for recognizing MHC class Ib, which involve NKG2D in both species [48,49,50]. Despite the differences in the ligand recognition mechanisms, the intracellular signaling mechanism involved in NK cell activation is conserved in these two species [48].

## 3. Changes in NK Cell Numbers and Subpopulations with Age

Several studies have characterized changes in the absolute number of circulating NK cells as well as the distribution of NK cell subsets with increasing age. Age does not appear to influence NK progenitor numbers in the peripheral blood or in the bone marrow [51]. Nonetheless, although there is some disagreement in the field, most studies report an increase in NK cell number with age (11 of 13 reported studies) (Table 1).

The markers used to label NK cells, or the gating strategy used to count them, may at least partially explain the discrepancy in the studies. The earliest studies relied on CD16 to distinguish NK cells from other lymphocytes, which may have resulted in an incomplete picture of total NK cell number. For example, Zang et al. used CD3 and CD16 expression to distinguish NK cells and reported no differences in peripheral NK cell numbers with age [69]. However, CD56 is now regarded as a more reliable and general marker of human NK cells than CD16. While CD16 is often expressed on the more mature subset, CD56^dim^ NK chambers, the CD56^bright^ NK cell subset sometimes lacks or expresses very low levels of CD16. It is possible that these earlier studies reported changes in CD56^dim^ NK cell populations. Later studies, which included CD56 to label NK cells, reported that absolute cell numbers increased with age (Table 1). With regards to changes in NK cell subpopulations, the studies listed in Table 1 almost unanimously report that there is a significant decrease in the percentage of CD56^bright^ cells (Table 1: six of six studies) [62,64,65,66,67,68] and a significant increase in CD56^dim^ cells (five of five studies) [62,65,66,67,68].

Unbiased approaches, such as single-cell RNA expression analysis, offer a possible approach to resolving some of the conflicting data on the effect of aging on NK cells. Transcriptional clustering has revealed two distinct signatures of splenic NK cells, which have a more active gene expression signature than NK cells present in the blood [70]. RNA-sequencing has confirmed the prevalence of several distinct subpopulations of NK cells in the human bone marrow and blood based on the expression of several conserved markers [71]. These findings expand upon the classical NK cell developmental model (CD56^bright^ → CD56^dim^CD57^−^ → CD56^dim^CD57^+^) and identify two subpopulations in the CD57^+^ population, distinguished by the expression of *CX3CR1* and *HAVCR2* to differentiate the terminally differentiated NK cell population from the matured NK population [71]. In addition, single-cell RNA expression analysis of NK cells isolated from peripheral blood from healthy subjects reveals even more heterogeneity among NK cells, with up to ten subsets of NK cells identified by unsupervised clustering, with conserved makers per cluster independent of individual variation or stimulation [72]. Further, Smith et al. identified three novel NK cell populations; CD56^neg^ type I interferon-responding NK cells have increased granzyme B mRNA in response to IL2 and a small population with low ribosomal expression [72]. Similar methods were employed to analyze specific gene expression variances of NK cells in the context of aging, which demonstrated an increase in expanded late low-cytotoxic subsets with enriched apoptotic signaling pathways and decreased virus defense responses, as compared to the younger group [73].

An important limitation to our insight into the effect of aging on NK cell phenotype is that the great majority of studies, especially in humans, have only investigated age-related changes in circulating NK. Dogra et al. have substantially advanced our understanding by examining NK cells from multiple anatomic sites and compartments of 60 autopsies of subjects across a wide range of ages (5–92 years) and ethnic backgrounds [74]. Notably, across donors, the dominance of CD56^bright^CD16^−^ vs. CD56^dim^CD16^+^ was tissue-specific, whether pooled or at the individual donor level. CD56^dim^CD16^+^ NK cells dominated in blood and in tissues rich in it, such as the bone marrow, spleen, and lungs. Moreover, CD56^dim^CD16^+^ NK cells were substantially more likely to express CD57 (indicative of terminal differentiation) in tissues where this subset is dominant [74]

In contrast with previous reports (Table 1: [62,65,66,67,68]), Dogra et al. found no relationship between the ratio between CD56^bright^CD16^−^ and CD56^dim^CD16^+^ NK cell subsets and age in blood or any other tissue. It will be important to understand why this one impressive study came to such a different result on this front.

The frequency of terminally differentiated CD56^dim^CD16^+^CD57^+^ NK cells nominally rose with age in blood, bone marrow, spleen, and lung, but the trend was only statistically significant in the bone marrow [74]. The lack of a clear age-related change in this subset in blood is broadly consistent with limited and inconsistent prior reports [64,66]. Age was also unrelated to the level of granzyme B expression by CD56^dim^CD16^+^ cells, nor the tissue-specific NK cell expression of the signaling adaptor FcεRIγ, an indicator of antibody-dependent cellular cytotoxicity (ADCC) potential [74]. Unlike its effects on T-cell differentiation and polyclonality (which clearly interfaces with aging, despite its uncertain effects on functional immune senescence [15]), cytomegalovirus serostatus did not impact NK cell frequency or tissue distribution [74].

Further investigation of age-related changes in the heterogeneity amongst NK cell populations may be vital in understanding the mechanism behind the loss of their functions with age.

## 4. Effect of Age on NK Cell Cytokine Production and Secretion

Decreased NK cell cytotoxic subtypes and function are associated with an increased risk of viral infections and cancers [75,76,77]. Dysfunctional NK cells are typically identified by decreased NK effector functions, such as impaired NKCC, as well as reduced IFN-γ secretion and expression of perforin and granzyme [78]. Release of cytokines, such as IFN-γ and TNF-α, activated the NK cells help coordinate a broader immune response, including the recruitment of macrophages [79] and neutrophils [80] at the site of cancer [81], infection [82], or senescent cells [83].

One potential cause of NK cell dysfunction with age may be changes in the systemic milieu [84,85]. Particularly notable is the well-established age-related decline in interleukin-2 (IL-2) production in humans [86]; IL-2 is a potent enhancer of both NKCC and NK cytokine secretion, such as TNF-α [87]. One study reported that TNF-α secretion is sustained in IL-2-stimulated NK cells from otherwise-healthy aging donors, despite their reduced IL-2 production [62]. Other studies have found that NK cells from older donors increase production of IFN-γ and/or TNF-α following IL-2 treatment [64,88], but to a lesser degree than that which occurs with younger donors [59,89]. Similar to IL-2, interleukin-12 (IL-12) enhances the cytotoxicity of NK cells and promotes IFN-γ secretion by NK cells [90,91]. Studies have reported a reduction in cytokines released by NK cells from older donors, despite activation with IL-2 [92,93,94], and chemokine production in older subjects, despite stimulation with IL-2 or IL-12 [95,96]. One caveat to these findings is that a reduction in cytokine or chemokine production per cell with age may possibly be attributable to the age-related shift from CD56^bright^ to CD56^dim^ NK cells, as the former are responsible for producing these mediators. Further investigation of age-related changes in the cytokine and chemokine production and secretion amongst NK cell populations may be vital in understanding the mechanism behind the decrease in NK cell activation and recruitment.

## 5. Changes in NK Cell Cytotoxic Activity with Age

A decline in NKCC with age has been well documented and has been linked to increased incidence of infectious diseases [73,97,98]. Age-related NKCC impairment has also been implicated in enabling the emergence of multiple noncommunicable diseases of aging [99]. For example, adult cancer incidence and mortality for all-site invasive cancers and for many individual cancer types rise progressively with age [100,101] and the decline in NK cell function with age is implicated in this increased risk [102]. NK cell surveillance provides a critical early defense against precancerous and cancerous cells [103,104]. Depressed NK cell activity in animals results in elevated tumor incidence [105] or increased metastasis [106,107]. Relatives of patients with familial melanoma [108] and individuals with a high family incidence of cancer [109] exhibit low NK cell activity compared to healthy controls; similarly, liver cirrhosis patients with reduced NKCC are prospectively at an elevated risk of progressing to liver cancer [110], and one study reported low NKCC to be longitudinally associated with a risk of cancer at all sites in the general Japanese population aged 40 and over [111]. A longitudinal study demonstrated a declining efficacy of NK cell cytotoxicity, coinciding with elevated risk of cancer in both males and females with age [84].

These epidemiological findings are supported by laboratory data. More than half of the studies (6 of 11 studies—Table 1) report a decrease with age in NK cell cytotoxicity against K562 cancer cells. The non-unanimous findings can be attributed to the lack in consistent NK cell markers, leading to potential contamination; in addition, some of the reports used IL-2, which augments the innate NKCC ability. One study [67] reported an age-related rise in expression of the inhibitory receptor KLRG1, particularly in CD56^dim^ NK cells; CD56^dim^ NK cells expressing high levels of KLRG1 had impaired cytotoxicity against the MCF-10A cancer line, which expresses high levels of KLRG1’s ligand E-cadherin, and silencing RNA against KLRG1 enhanced these cells’ NKCC against this target cell type. Notably, however, one study reported an increase in cytotoxic activity with age, associated with a shift from the CD56^+^57^−^ to the CD56^+^57^+^ phenotype [54]. The impact of aging on NK cell toxicity against senescent cells remains to be investigated. Identifying the cause(s) for the age-related decline in NKCC is a necessary step toward identifying interventions to retard or reverse this decline, and thus potentially mitigate its contribution to age-related diseases.

NK cell killing of target cells is dependent on the NK cell receptors necessary for target cell recognition. Age-related declines in the percentage of NK cells expressing the activating receptors, NKp30 or NKp46 [66,112,113], have been reported. One study reported an age-related increase in NK cells expressing the inhibitory receptor KLRG1, particularly but not only in the CD56^dim^ subset [67]; contrarily, another group reported a decline in both NKG2A and KLRG1 [64]. The percentage of NK cells expressing inhibitory KIRs has been reported to increase with age [54]. Taken together, these results point towards an alteration in the ability to receive pro-activation signaling from the target cells, possibly indicating an overall shift towards a more inhibitory phenotype.

As a possible indication of the functional significance of such changes, researchers have investigated diversity and frequency of KIR genotypes in otherwise-healthy aging persons [114] and centenarians [115] as compared to younger controls. The first study did not reveal a clear enrichment of either KIR diversity or particular KIR genes in older compared to younger Irish subjects, suggesting that neither confers a late-life survivorship advantage [114]. The analysis of the HLA-DRB1 and KIR receptors/HLA ligand frequencies in centenarians and controls from Sicily also demonstrated no significant difference in KIR receptors [115].

A reduction in NK cell perforin expression has been reported to be the mechanism behind age-dependent decline in NK cell cytotoxic function [116]. However, one study found no significant differences in the perforin content, distribution, and utilization in the lytic assays in NK cells from young (≤35-year-old) and old (≥61-year-old) groups, but evidence of impaired perforin release through degranulation upon coculture with target cells [113]. Certain compounds, such as ionomycin and phorbol myristate acetate (PMA), are used in vitro to activate NK cells, bypassing the need for target cell surface receptors to induce NK cell degranulation [99]. NK cells from older donors maintain their sensitivity to this assay, suggesting that the age-related decline in NKCC may be due to loss of these discussed receptors and/or to cell signaling, and not the ability to produce cytotoxic granules when activated [88,89,99].

These results collectively indicate that NK cells from older subjects have undergone biological changes, which impact their ability to interact with other immune cells and their target cells. One hypothesis to explain the increase in the numbers of cytotoxic NK cells with age may be that it is an attempt to compensate for the age-related loss in effector function at the individual NK cell level.

## 6. Effect of Age on NK Cell Signal Transduction

While it has become clear that NK cell function declines with age, few studies provide clues about the underlying causes. The process of NK cell cytotoxicity is a complex and tightly coordinated series of events, which ultimately result in the release of lytic granules into the immunological synapse (IS) between the NK cell and a bound target cell (Figure 1) [117]. Several key checkpoints in this pathway involve the turnover of phosphoinositide molecules by phospholipase C (PLC), calcium mobilization and influx, polarization of secretory lysosomes to the IS, and fusion of these vesicles with the NK cell membrane [117,118,119,120]. Ca^2+^ release is also required for critical degranulation checkpoints, such as migration of the mitochondria to the IS (which controls subsequent Ca^2+^ influx) and fusion of secretory lysosomes with the NK cell membrane [121,122].

The second messenger, phosphoinositol phosphate 3 (IP3), is critical to this process, as it stimulates the release of internal Ca^2+^ stores, which leads to downstream activation events. The IP3 production, following exposure to K562 cancer cells, was significantly reduced in NK cells from older subjects [61]. As noted previously, one study reported that the Ca^2+^ mobilization in NK cells from older donors is reduced, relative to the level observed in NK cells from younger donors in response to IL-2 [62].The decline in the mobilization of Ca^2+^ resulting from the impaired production of IP3 might therefore contribute to the impaired cytotoxicity of the NK observed in older subjects [118].

In addition to the decreased Ca^2+^ mobilization, NK cells from older individuals exhibit diminished perforin expression and release, which likely contributes to the age-dependent decline in NK cell toxicity [113,116]. One study measured the abundance of intracellular perforin in unstimulated CD56^+^CD16^+^ NK cells from various age groups and demonstrated an age-dependent decline in the percentage of perforin-positive NK cells in individuals that are 70 years and older, which was only partially remedied by treatment with IL-2 [116]. In another study with a much larger sample size, no decline in the expression of perforin with age was observed nor were changes in the fusion of secretory lysosomes with the NK cell membrane [113]. However, NK cells from older donors in this study did exhibit diminished perforin binding to the surface of K562 cancer cells [113]. As noted previously, the decline in IL-2 production with aging is well-known in humans [86]; thus, it is possible that in vitro experiments using activated NK cells could hide any difference in resting perforin expression between young and old NK cells [127].

## 7. NK Cell Exhaustion

The phenomenon of cell exhaustion is well known and characterized in T cells; however, it is not known whether NK cells undergo a similar state of exhaustion [128,129]. The impairment of the cytotoxic activity of NK cells in cancer patients is well known. For instance, a significant reduction in the numbers of NK cells expressing NKG2D, NKp30, NKp46, and perforin was observed in patients suffering from pancreatic, gastric, and colorectal cancers [130]. A similar loss in cytolytic function of NK cells is also observed in patients suffering from chronic viral infection [131]. An investigation of the molecular signaling in these NK cells may shed new light on any potential exhaustion phenotype in NK cells and help to distinguish them from senescent NK cells.

While NK cells themselves are short lived (around two weeks), human NK cells exhibit telomere shortening and a decrease in telomerase activity with age [132,133]. Cellular differentiation has a role in telomere shortening, leading to the more mature CD56^bright^ NK cells having shorter relative telomere length than the immature CD56^dim^ subset [132,134], but all subsets of NK cells have decreased telomere length with age, with CD56^bright^ cells demonstrating the greatest decrease [132]. Interestingly, NK cells expressing the activation markers, NKG2D and LFA-1, have shorter telomeres as compared to those without, while those expressing inhibitory markers, such as KIR or NKG2A, had an extremely heterogenous telomere length with no real pattern [134]. Taken together, these studies point to telomere attrition as a potential factor for diminished NK cell function with age, but more research would need to be conducted in order to determine any actual functional effects on cytotoxicity.

## 8. Impact of the Aging Systemic Milieu on NK Cells

To this point we have largely focused on predominantly cell-autonomous changes in NK cells with age, but changes in the local microenvironment during NK cell maturation or locally at the site of abnormal cells during aging may also skew NK cell behavior. Pro-inflammatory and anti-inflammatory cytokines work together to mediate proper immune responses and are vital to the healthy aging of an individual [135]. Chronic sterile systemic inflammation is often considered emblematic of dysfunction associated with aging, and is a powerful risk factor for morbidity and mortality in age-related diseases, such as sarcopenia [136], obesity [137], and Alzheimer’s [138]. Following transplantation into old mice, NK cells isolated from young mice exhibit significant impairment in NK cell function in ex vivo assays, whereas the transfer of NK cells from old mice to young mice seems to fully restore their cytotoxic potential [139]. These findings suggest a strong influence of host tissue microenvironment on NK cell function in aging and are consistent with in vitro stimulation experiments of human NK cells [64,88]. However, the interpretation of studies in aging mice must be taken with caution, as mouse NK cells are functionally quite different from those of humans at the protein level (mouse have much lower perforin and granzyme levels) and in terms of localization patterns [140].

The aged environment might lack the stimulating factors necessary to achieve maximal NK function. With age, the levels of IL-2 [86] and IL-15 [141], vital cytokines for NK cell development and survival, decline, which may contribute to the decline in NK cell surveillance with age. In parallel, the levels of inflammatory cytokines, such as IL-6 [142] and GDF15 [143,144], increase with age, contributing to the aged inflammatory microenvironment. Treatment of NK cells in culture with interleukins 2, 12, or 15 (IL-2, IL-12, IL-15) and interferon-α (IFN-α) can increase their cytotoxicity towards cancer cells and even toward cancer lines that are generally resistant to NK cell killing [145,146].

Studies are mixed on whether stimulation of aged NK cells with IL-2 ex vivo is sufficient to restore NKCC and related activities. One study reported full recovery of NKCC following IL-2 stimulation of in NK cells from old (60–80 years old) and even very old (80–100) vs. middle-aged (18–60) subjects [89]. However, others report that although NK cell activation by cytokine treatment does improve cytotoxicity in old-derived NK cells, it never reaches the level observed in NK cells derived from young donors, indicating a disruption in cell-level activation signaling with age [64,147]. One study demonstrated a decreased sensitivity of NK cells to IL-2 when derived from older donors [62], which may at least in part be explained by the use of the SENIEUR patient selection criteria in this study, which attempted to avoid confounding factors that elicit strong immunological responses, such as current infection, cancer, recent myocardial infarction or stroke, lymphoproliferative disorders, or laboratory findings outside the reference values for the subjects’ age [148].

The evidence is similarly mixed on whether IL-2 stimulation of aged NK cells is sufficient to normalize production and secretion of NKCC effectors. One study reported an age-dependent decline in the percentage of perforin-positive NK cells derived from individuals that are 70 years and older, which was only partially remedied by treatment with IL-2 [116]. However, in another study with a much larger sample size, no decline in the expression of perforin with age was observed, nor were changes in the fusion of secretory lysosomes with the NK cell membrane; however, they did demonstrate diminished perforin binding to the surface of K562 cancer cells [113]. As the decline in IL-2 production with aging is well known in humans [86], it is possible that in vitro experiments using activated NK cells could hide any difference in resting perforin expression between young and old NK cells [127].

Epigenetics makers are known to be heavily influenced as organisms age. While there not many studies on the role of epigenetic modifications on the differentiation and function of NK cells, especially not in relation to age, it is very clear that the activation state of NK cells is epigenetically regulated [149]. NK cell activity is modulated by the complex interaction of activating and inhibiting receptors, and how much they can be activated seem to be modulated by epigenetics [150]. The exposure to cytokines, such as IL-6 or INF-γ, can influence NK cell activation; however, the strength of the response is modified by prior exposure to other stimuli, which alter the epigenetic state of the promotor for their receptors [151]. Exposure to factors that are known to influence systemic aging—such as CMV infection [152], exercise [153], and emotional stress [154], have also been demonstrated to change the epigenetic status of NK cells. More broadly speaking, long-term exposures to a variety of stresses lead to epigenetic modifications [149,152]

Interestingly, it seems that the differentiation of CD56^dim^ NK cells affects epigenetic clocks, which use the level and pattern of DNA methylation in cells either in blood or other tissues to predict biological age and mortality [155]. Further, Jonkman et al. reported that cytotoxic NK cells as well as T-cells seem to be the important drivers of these clocks, and that blood CD56^dim^ and CD57^bright^ NK cells from the same donor are assessed by several such clocks to be biologically much older than CD56^bright^ cells [155]. Certainly, our understanding of the role of the interaction of aged NK cells with the systemic milieu is incomplete, suggesting that additional cell-autonomous or microenvironmental factors remain to be identified.

## 9. Neuroendocrine Signaling and NK Cell Aging

Neuroendocrine factors affect NK cells, and many studies have demonstrated the influence of stress [156] and the nervous system on NK cell activity [157,158,159]. Some of these factors are known to change with age: for instance, glucocorticoid (GC) levels increase with age in human serum [160], even in relatively healthy aging people, as assessed by the SENIEUR criteria [161], and GCs were demonstrated to have an inhibitory effect on NK cell functions [162]. This effect is due to decreased gene expression for essential genes, such as perforin and granzyme B, leading to an overall reduction in cytotoxicity [116] and IFN-γ production [163].

The serotonin receptor agonist, quipazine, enhanced NK cell function, whereas various dopamine/serotonin antagonists inhibited CD16-mediated NK cell function [112]. In addition, NK cells express very high levels of dopamine receptors, and their activation seems to have an inhibitory function on human NK cells [164]. Similarly, epinephrine and norepinephrine seem to primarily inhibit NK cell cytotoxicity and cytokine production [165,166].

Neuropeptides present in the peripheral blood bind to and can modulate NK cell activity [167]. Galanin modulates IFN-γ secretion and sensitizes NK cells towards specific cytokines [168], while neuropeptide substance P increases the cytotoxicity of NK cells and helps with NK cell migration [168,169]. Both galanin and neuropeptide substance P decrease expression with age [170,171], suggesting that the observed decline with age may contribute to the age-related impairment in NK function.

Insulin-like growth factor 1 (IGF-1) is an important growth factor that promotes NK cell development from CD34+ cells and increases NK cell cytotoxicity [172]. IGF-1 levels decline with age [173,174]. Additionally, the secretome of senescent cells contains several pro-inflammatory factors, including TNF-α [175], which has been demonstrated to induce a state of IGF resistance [176]. Combined, the reduction in circulating IGF-1 and tissue resistance to IGF-1 signaling may contribute to the age-related changes to NK cells and the ensuing decrease in senescent cell clearing.

Other growth factors, such as platelet-derived growth factors D and DD (PDGF-D and PDGF-DD), are able to interact with the surface of NK cells to induce NK cell survival [177] and cytokine secretion [178]. There are conflicting results on the age-induced changes to levels of PDGF, with reports of decreased [179], unchanged [180], and increased [181] PDGF levels with age. These studies focus on PDGF in general rather than the PDGF-D and PDGF-DD specifically, and thus these levels would need to be examined in the context of age to further understand their role in NK cell aging.

In contrast to IGF-1 and PDGF, transforming growth factor beta (TGF-β) has been demonstrated to inhibit NK cell development and function [182,183]. TGF-β has been demonstrated to block IL-15-induced metabolic activity and proliferation of NK cells by inhibiting the mechanistic target of rapamycin (mTOR) activity [184] and the expression of NKp30 [185] and NKG2D receptors [186]. Senescent fibroblasts secrete TGF- β [187], and plasma TGF- β levels have been demonstrated to increase with age [188].

## 10. Reducing the Impacts of Secondary Aging on NK Cell Function

Ewald W. Busse is credited with introducing the conceptual distinction between primary aging, denoting age changes driven by intrinsic metabolic processes, and secondary aging, driven by supernumerary cellular and molecular damage inflicted through lifestyle and environmental exposures [189,190]. Recent years have observed rapid advances in the biology of aging, leading to the discovery, development, and translation from animal models of candidate interventions targeting primary aging processes, but drivers of secondary aging are already amenable to intervention today [191,192].

### 10.1. Micronutrient Deficiency

One recognized source of secondary aging in older persons is micronutrient deficiencies, resulting from some combination of malabsorption, low intake, and an increase in the aging body’s requirements [193,194,195,196,197]. A dysregulation of essential nutrients, such as vitamin B_6_, vitamin B_12_, folic acid, and zinc, which are essential for cell-mediated immunity [198,199,200], are linked to various age-related pathologies and diseases, including inflammation [201,202], bone aging [203] and osteoporosis [204], and Alzheimer’s disease [205,206], in addition to a decrease in NK cell count and/or cytotoxicity [197,207,208,209].

Nutrient supplementation studies demonstrate significant promise in mitigating these secondary aging effects. Studies have demonstrated that the age-related decrease in zinc levels correlate with a reduction in NK cell cytotoxicity [210]. Zinc supplementation in NK cell cultures improves the age-related decline in proliferation/differentiation of CD34^+^ cells, in part by restoring the transcription factor Gata-3 [51]. NK cell cytotoxicity in healthy elderly subjects is reported to be partially restored through supplementation with zinc [210,211].

Similarly, vitamin B_12_ and folate (vitamin B9) dysregulation have been correlated with lower NK cell counts and activity, and can be improved through supplementation, which increases NK cell counts and activity in aging rats [200,209]. Again, a nutritional formula containing 120 IU vitamin E, 3.8 µg vitamin B_12_, 400 µg folic acid, and other nutrients increased the NK activity relative to the placebo in healthy subjects aged 70 years and older [212].

To our knowledge, the effect of vitamin D supplementation on NK cell function has not yet been tested in studies in aging humans, but the mechanisms of action of vitamin D in immunity are thought to principally involve the innate immune system [213]. A meta-analysis of individual participant data (IPD) from randomized controlled trials of vitamin D supplementation in subjects from birth up to 95 years of age reported a reduced risk of acute respiratory tract infections, particularly in trials involving daily or weekly vitamin D and in subjects with baseline deficient 25-hydroxyvitamin D levels [213]. Inconclusive data suggest that vitamin D deficiency may increase risk of infection with SARS-CoV-2 and/or adverse outcomes in COVID-19 [214], and evidence supports a potential role for vitamin D repletion in the prevention and amelioration of acute respiratory distress syndrome [215,216]. NK cell numbers and/or cytolytic activity were positively associated with serum vitamin D levels in a cross-sectional study in otherwise-healthy nonagenarians and centenarians [217].

In addition, the effect of supplementation with other nutrients whose deficiency is implicated in age-related immune dysfunction, such as vitamin E [218,219] on NK cell activity in aging subjects, remains to be investigated.

### 10.2. Stress Reduction

As noted above, GC levels rise with age [160] and are further elevated in association with elevated psychological stress [161] and impaired NK cell functions [162]. Consistent with this, extensive evidence has demonstrated that subjective stress levels and stressful life events predict impaired immunity [220], including increased susceptibility to induced rhinovirus infection [221] and less durable antibody titers after vaccination [222]. The increase in influenza-specific IFN-γ production following influenza vaccination and following four weeks of aerobic exercise appeared to be partially mediated by psychosocial factors [223]. Again, we are unaware of human challenge studies testing the effects of induced or self-reported stress on NK activity, but a study in mice subjected to selective REM sleep deprivation found that the ensuing impairment in NK cell numbers and activity was mediated by an upregulation in NK cell β–adrenergic receptors, possibly caused by an increase in GC levels, which was itself caused by the sleep restriction protocol [224].

A potential approach to reducing secondary aging of the immune system (including possibly NK function) in aging is therefore intervention against subjective stress and the associated elevation in GC levels. Modest evidence suggests that such modalities, as cognitive, psychological, or exercise interventions to lower chronic stress, may improve immune function [225]. Magnesium supplementation in older subjects has also been reported to lower nighttime GC levels [226,227].

### 10.3. Sleep Health and Circadian Alignment

The quality and quantity of sleep and the amplitude of circadian rhythms decline with age [228], and low objectively measured sleep quantity and/or quality are linked to neurodegenerative [229] and other diseases of aging [230,231], as well as total mortality [232,233,234]. Insufficient or circadian-misaligned sleep are known to cause impaired immune function [235,236]. A single night of imposed total [237] or even partial [238,239] sleep restriction, or nights of shorter vs. longer self-reported sleep [240,241] are followed by lower NK cell number and/or functions in humans the following day. Conversely, a night of recovery sleep allows NK activity to normalize [238,239], although IL-2 levels remained suppressed [238].

Consistent with this, animal studies demonstrate that chronic partial sleep restriction accelerates the growth and metastasis of experimental cancer associated with a decrease in tumor dendritic, NK, and T-cells [242,243], while better sleep is associated with longer survival in human cancer patients [244,245].

Therefore, the means to improve sleep quality or quantity, or to entrain circadian rhythms, may be a potential means to ameliorate secondary immunological aging and age impairments in NK function. In addition to cognitive-behavioral therapy and therapeutic sleep restriction for insomnia and sleep hygiene for more general poor sleep [246], some specific interventions demonstrated to improve sleep in older persons include magnesium supplementation [226,227] and creating a peripheral temperature gradient via wearing socks in bed [247,248] or through pre-bed hot baths [249].

Tobacco smoking and obesity also suppress NK cell activity, which reverses upon smoking cessation or weight loss, respectively, and should be pursued if the patient can be engaged [250].

## 11. Harnessing NK Cells as Immunosenolytics

Senescent cells are cells that have undergone an irreversible cell-cycle arrest in response to a range of stressors or as part of physiological processes, and they are known to secrete a pleiotropy of inflammatory, growth, and matrix-degrading factors, collectively termed the senescence-associated phenotype (SASP) [13,251]. By restricting tissue renewal with age and through promotion of inflammation, senescent cells have been implicated as drivers of the degenerative aging processes and contribute to specific diseases of aging [14].

Senolytics (drugs that selectively remove senescent cells from the body) have been demonstrated to increase the health span in aging laboratory animals and are advancing into clinical trials for human translation [10,252]. However, they bear a significant potential limitation. Such drugs rely on the differential activity of essential cell metabolic pathways, including anti-apoptotic, pro-survival, mitochondrial metabolic, kinase [83,253], and glutaminolytic pathways [254]. Such mechanisms of action necessarily perturb the healthy function of non-senescent cells, raising the specter of important side-effects, despite the overall positive effects on the health span and lifespan observed in animal models of aging and its diseases. A more physiological strategy that exploits the intrinsic immunosurveillance of senescent cells might offer greater potential to favorably impact aging and diseases of aging.

Recently, the discovery of NK cell-mediated immune surveillance of senescent cells has led to an interest in harnessing this effector function as an approach to combating aging and age-related diseases [10,11,12]. As noted previously, NK cells are the chief lymphocytes responsible for the clearance of senescent cells during wound resolution, development, and other physiological processes, and also from aging or diseased tissues. Thus, understanding the causes of the NK cell functional decline with age may allow us to develop therapeutic strategies to rejuvenate or fortify NK-mediated immunosurveillance of senescent cells.

One potential approach would be to reverse age-related deficits in the intrinsic capacities of this process. As we have discussed, NK function declines with age, and although the impact of aging on NK cell toxicity against senescent cells has not yet been elucidated, the known mechanisms driving and enforcing the decline in other NK cell functions may present targets for restoring the immunosurveillance of cancerous, virally infected, and senescent cells.

A second potential immunosurveillance-enhancing approach would be to target mechanisms whereby senescent cells evade immunological clearance. Such mechanisms include the shedding of NK binding ligands from the senescent cell surface [255] by matrix metalloproteinases (which are part of the SASP) [256] and upregulation of the inhibitory receptor HLA-E [257]. Although chronic inhibition of these mechanisms may be expected to lead to excessive or off-target removal of senescent or other cells, a more transient approach, similar to that which has been employed in most preclinical studies of senolytic drugs, may open a favorable therapeutic window.

A third approach would be to fortify the endogenous armamentarium of the immune system via transfer therapy of expanded and potentially engineered NK cells. Due to their significant involvement in the immune surveillance of cancer cells, NK cells have recently become a promising tool in cancer immunotherapy [258]. Several advancements have been made to enhance the efficacy of NK cells in response to the tumor microenvironment inhibition of NK cell activity [259], including the chimeric antigen receptor (CAR)-NK cell production and ex vivo activation of NK cells by cytokines [259,260].

CAR-T cells targeting a candidate senescent cell surface marker were reported to extend survival in a murine model of post-chemotherapy senescent cell accumulation, and to partially normalize liver fibrosis resulting from either treatment with CCl4 or from diet-induced nonalcoholic steatohepatitis (NASH) [261]. Similar strategies could be exploited to engineer NK cells to target senescent cells for elimination from tissues. NK cells, rather than T-cells, are the principal actors in the immune surveillance of senescent cells specifically, but even in the cancer context, CAR-NK cells offer a number of potential advantages over CAR-T therapy [262,263]. First, the profile of cytokines released by NK cells is thought to be substantially less likely to lead to a systemic inflammatory response; such a “cytokine release syndrome” is a critical side-effect of CAR-T therapy. The risk is further reduced by less expansion of CAR-NK cells after the infusion into patients. Additionally, CAR-NK cells do not persist as long in the body as CAR-T cells after mounting an attack on tumor cells in vivo, which lowers the risk of graft vs. host disease and malignant transformation. While CAR-T cells lose their cytotoxic capacity if expression of the CAR system is lost or its target antigen is not present on subpopulations of target cells, CAR-NK, regardless, would retain their native receptors as a backup system for targeting aberrant cells.

## 12. Perspective and Conclusions

NK cells play critical roles in combating cancer, disease, senescence, and infection, yet their function appears to decline with age. As part of immunosenescence, decline in NK cell cytotoxicity is likely a major cause of the increased susceptibility to viral infection, increased overall disease burden, and accumulation of senescent and cancer cells observed in the elderly [264]. Considering this, a growing number of studies have sought to probe specific changes that occur in NK cell biology and functionality with aging. Uncovering changes with age in gene expression, marker expression/distribution, degranulation, and internal or systemic signaling can lead to a better understanding of how NK cells age and the impact of their local environment. These insights can aid in the development of tailored rejuvenation and intervention strategies, which may restore the functionality of native NK cells or enable the augmentation of their forces with allogenic or autologus cells.

A recent study using mass cytometry (also known as or cytometry by time-of-flight, or CyTOF) has uncovered between 6000 and 30,000 distinct NK cell phenotypes within a given individual based on unique combinations of 35 cell surface antigens, whose expression is determined by both by genetics and the environment [265]. A further characterization of surface antigens and phenotypic expression changes with age will be important to developing better therapeutic interventions for improving NK cell functions for age-related diseases.

The transplantation of allogenic NK cells has demonstrated great promise in cancer treatment and as such, demonstrate promise for the transplantation of young allogenic or cord-blood-derived NK cells for treatment of various age-related diseases and aging in general [266,267]. Furthermore, CAR-T cells have proven to be powerful tools for immunotherapy for the treatment of cancer [268,269] and, recently, CAR-NK cell have emerged as an attractive alternative [263,268]. In recent clinical trials, CAR-NK cell immunotherapy has not only been found to be effective, it has also significantly improved affordability (by using off-the-shelf NK cells), reduced sensitivity to immune checkpoints, and lowered the probability of nonspecific neurotoxicity, as compared to CAR-T therapy [270,271].

In sum, much is known about the changes with age in NK cell function, but much less about the factors driving it. New tools and technologies will increasingly enable the understanding and therapeutic intervention of NK cell aging, allowing people of increasingly advanced ages to live lives more free of infection, cancer, senescence, and ultimately advance the widely emerging revolution in longevity therapeutics.

## Figures and Tables

**Figure 1 cells-11-01017-f001:**
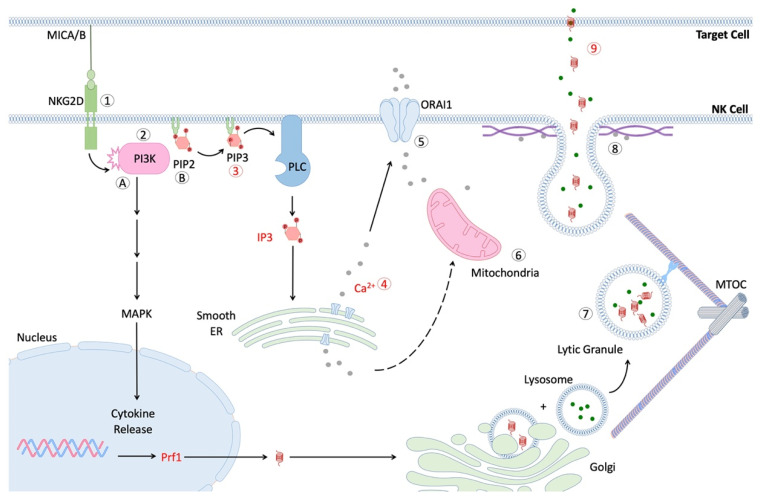
Scheme of NK cell signaling after formation of immunological synapse (IS). Interaction of activating receptor with cognate ligand leads to phosphorylation of cytoplasmic tail and recruitment of PI3K [61,120]. Pathway (**A**) activation of PI3K and PLC lead to downstream signaling via MAPK and NFκB pathways, respectively, thus facilitating the production and secretion of cytokines, such as IFN-γ. Activation of MAPK signaling also contributes to MTOC migration and the polarization of lytic granules. Pathway (**B**). Conversion of phosphatidylinositol-4, 5-bisphosphate (PIP2) to phosphatidylinositol-3, 4, 5-triphosphate (PIP3) allows for subsequent generation of the secondary messenger inositol trisphosphate (IP3) via phospholipase C (PLC) [117]. IP3 contributes to the release of internal calcium stores from the smooth endoplasmic reticulum [123], which in turn facilitates mitochondrial migration to IS and influx of extracellular calcium via the ORAI1 transporter [118,119]. Additionally, the microtubule organization center (MTOC) migrates towards the IS and allows for efficient transport of lytic granules (secretory lysosomes) to travel to the IS [124,125]. Lastly, lytic granules fuse with the NK cell membrane in a calcium-dependent manner and release perforin and granzymes into the IS [126]. These proteins form holes in the target cell membrane and induce apoptosis.

**Table 1 cells-11-01017-t001:** Changes in natural killer cell phenotype, subset, and cytotoxicity throughout the aging process. Summary of age-related changes in natural killer cell phenotype, subset distribution, and cytotoxicity. Age-related changes in natural killer cell phenotype, subset distribution, and cytotoxicity against cancer cell lines.

Young Donor Age Range	Old Donor Age Range	NK Cell Definition	NK Numbers	Aged Phenotype	Target Cell	Cytotoxic Potential	Assay	IL-2 Activation	Ref
25–34	75–84	*CD16^+^/Leu7^±^	Increased with age	CD16^+^ and Leu7^+^ increased	-	-	-	-	[52]
21–38	71–97	*Leu7^+^/Leu11a^±^	Increased with age	-	K562	Decreased cytotoxic potential with age in both subsets of NK cells	Cr Release Assay	-	[53]
20–60	80+	CD16^+^ CD56^+^ CD5^±^	-	CD16^+^ CD57^+^ increased; CD56^+^/CD57^−^ decreased	K562	Increase in cytotoxic potential with age	Cr Release Assay	-	[54]
27 ± 6	81 ± 7	CD16^+^	Increased with age	-	K562	Similar binding capacity, Decreased cytotoxicity with age	Flow Cytometry	-	[55]
19–36; 50–68	100–106	CD16^+^ CD57^±^	Increased with age	CD16^+^ CD57^−^ increased; no change in CD16^+^ CD57^+^	K562	No significant difference between age groups	Cr Release Assay	-	[56]
25–35	75–94	CD16^+^ CD56^+^ KIRDL^+^ or CD16^+^ CD56^+^ KIR2DL3^−^	Increased with age	CD16^+^, CD56^+^ and GL138^+^ increased	K562	Decrease in lytic activity of CD16^+^ cells	Cr Release Assay	Yes	[57]
23–35	65–100	CD3^−^ CD56^+^	No change	-	K562	No significant difference between age groups	Cr Release Assay	Yes	[58]
23–35	67–95	CD3^−^ CD56^+^	No change	-	K562	Decrease in cytotoxic with age	Cr Release Assay	Yes	[59]
30 ± 2	85 ± 2	CD16^+^	Increased with age	-	K562	Decrease in cytotoxic potential with age	Cr Release Assay	-	[60]
30 ± 2	85 ± 2	CD3^−^ CD16^+^ CD56^+^	Increased with age	-	K562	Decrease in cytotoxic potential with age	Cr Release Assay	-	[61]
19–39	77–89	CD3^−^ CD56^dim^ or CD3^−^ CD56^bright^	Increased with age	CD56^dim^ Increased; CD56^bright^ decreased	-	-	-	-	[62]
21–30	65+	CD3^−^ CD56^+^	-	CD94 and NKG2A decreased; KIR increased	P815	CD16 mediated cytotoxicity did not vary with age	Cr Release Assay	Yes	[63]
Cord blood; Young <60	60–75; 75–80	CD3^−^ CD56^dim^ or CD3^−^ CD56^bright^	Increased in very old	CD56^bright^ decreased; NKp30 decreased; NKp46 decreased; KIRs increased	K562	No significant difference between age groups	Cr Release Assay	Yes	[64]
<60	60+	CD3^−^ CD56^dim^ or CD3^−^ CD56^bright^	-	CD56^dim^ increased; CD56^bright^ decreased; NKG2A decreased; KLRG1 increased	-	-	-	-	[65]
Children <18; adults 19–59	60+	CD3^−^ CD56^dim^ or CD3^−^ CD56^bright^	Increased in elderly, no difference between children and adults	CD56^dim^ increased, CD56^bright^ decreased, NKp30 and NKp46 decreased with age	K562	No significant difference between age groups	Flow Cytometry	Yes	[66]
20–34	70–86	CD16^+^ CD56^dim^ or CD16^+^ CD56^bright^	-	CD56^dim^ increased, CD56^bright^ decreased, KLRG1 increased with age	MCF-10A	-	-	Yes	[67]
41–50, 51–60	61–70, 71–80	CD3^−^ CD56^dim^ or CD3^−^ CD56^bright^	Increased with age	CD56^dim^ increased, CD56^bright^ decreased	-	-	-	Yes	[68]

* Leu7 = CD57 & Leu11a = CD16.

## Data Availability

Data are available on request.

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
