# Peer review of "Aging of the Immune System: Focus on Natural Killer Cells Phenotype and Functions"

_cells, 2022, doi:10.3390/cells11061017_

Round 1

Reviewer 1 Report

The review article of Brauning A et al. on the age-related changes in NK cells addresses with one of the most interesting topic in immunology and in cancer biology and therapy.

The authors cover numerous aspects related to the role and characteristics of NK by collecting extensive information from the literature, many of which refer to very old works.

The topic is not new and is the subject of other recent reviews. Authors should include, integrate and comment in the text of this manuscript some of the more interesting reviews, such as Dogra P et al. Cell 2020, Salminen A. Aging Research Reviews 2021.

An aspect that is missing from the review is that relating to the role of epigenetic modifications in NK with aging (see and include for example the work of Lu J et al. Aging Cell 2020).

Some other changes required:

  • Lines 61-62: please, revise the phrase! "secondary lymph nodes" should have been "secondary lymphoid organs" instead.
  • Lines 138-141: see Dogra P et al. Cell 2020.
  • Lines 143-151: authors should provide some details on the different signatures and/or subsets.
  • Line 144: the reference is inappropriate.
  • Figure 1: in the description, the pathways A and B are inverted.

Author Response

Reviewer 1:

The review article of Brauning A et al. on the age-related changes in NK cells addresses with one of the most interesting topic in immunology and in cancer biology and therapy.
The authors cover numerous aspects related to the role and characteristics of NK by collecting extensive information from the literature, many of which refer to very old works.

The topic is not new and is the subject of other recent reviews. Authors should include, integrate and comment in the text of this manuscript some of the more interesting reviews, such as Dogra P et al. Cell 2020, Salminen A. Aging Research Reviews 2021.
Response: We want to thank the reviewer for suggesting these references. We have made substantial edits in our review to include those. Please find relevant changes as follows: 

Dogra P et al. Cell 2020 has been integrated and used to expand upon the review by commenting on the distribution of NK cells and subpopulations with age throughout the body. Previously the table and review focused on PBMC isolated NK cells but the addition of this paper widens the scope of the review. Please see changes made in the section entitled ‘Changes in NK cell numbers and subpopulations with age’ line numbers This was added to lines 163-186.

Additionally, mention of the Salminen A. Aging Research Reviews 2021 has been added to the section regarding neuroendocrine factors and NK cells. This review highlights the effects of SASP factors on the feed-forward progression and immunosuppression. This information ties into the effects of secreted cytokines and molecules on growth factors and thus NK cells. Please find the relevant edits in the section ‘Neuroendocrine signaling and NK cell aging’ in lines The review was added to lines 447-466.

An aspect that is missing from the review is that relating to the role of epigenetic modifications in NK with aging (see and include for example the work of Lu J et al. Aging Cell 2020).
Response: We like to thank the reviewer for this comment/suggestion and we have included references to changes in the epigenetic signature of NK cells and their impact on aging in the section ‘Impact of the aging systemic milieu on NK cells in the lines 405-422. However, we were unable to find the citation for Lu J et al. Aging Cell 2020.

Some other changes required:

Lines 61-62: please, revise the phrase! "secondary lymph nodes" should have been "secondary lymphoid organs" instead.
Response: We agree with this comment and have made the change from "secondary lymph nodes" to "secondary lymphoid organs" in line 63.

Lines 138-141: see Dogra P et al. Cell 2020.

Response: We have added the findings from Dogra P et al. Cell 2020 to expand upon the review by commenting on the distribution of NK cells and subpopulations with age throughout the body. Previously the table and review focused on PBMC isolated NK cells but the addition of this paper widens the scope of the review. This was added to lines 163-186.

Lines 143-151: authors should provide some details on the different signatures and/or subsets.

Response: Thank you for giving us an opportunity to clarify this. Details on the different subsets and their identifying features were added to these cited lines at lines 149-152 and 156-159.

Line 144: the reference is inappropriate.

Response: We have corrected this reference has been updated to the correct citation from Crinier et al. Immunity 2018 please see the line number 146 in the revised manuscript.

Figure 1: in the description, the pathways A and B are inverted.

Response: Thank you for the opportunity to make this correction. We have updated the Figure legend to correctly describe the pathways in the diagram.

Reviewer 2 Report

This is a comprehensive review of a highly interesting and topical research area and I learnt a lot. The review is accompanied by a colourful figure and an informative table documenting the studies which associate ageing with increasing NK cell dysfunction. The sections are appropriate and introduced in a logical fashion that helps the flow of interesting information.

I have no major comments. However, given that NK cell receptors, such as NKp44, have been shown to recognise growth factors like PDGF-D (PMID: 29275861 & PMID: 35027451) and the gene for NKp44 (NCR2) has been associated with telomere length (PMID 29151059), I'd be very interested to know the authors thoughts and think it might improve the review further if the authors could comment or speculate on the possible role of secreted factors, such as growth factors, on NK cell immunosenescence and ageing?

Author Response

Reviewer 2:
Given that NK cell receptors, such as NKp44, have been shown to recognize growth factors like PDGF-D (PMID: 29275861 & PMID: 35027451) and the gene for NKp44 (NCR2) has been associated with telomere length (PMID 29151059), I'd be very interested to know the authors thoughts and think it might improve the review further if the authors could comment or speculate on the possible role of secreted factors, such as growth factors, on NK cell immunosenescence and ageing?
Response: We want to thank the reviewer for this suggestion we have made substantial edits to include the following sections in the revised version. More specifically, we have included a discussion of the effect of growth factors on the aging of NK cells in our section ‘Neuroendocrine signaling and NK cell aging’. The section on neuroendocrine signaling and NK cell aging was added to describe the role of PDGF-D along with other growth factors (IGF-1 & TGF-beta) on NK cells and the effect of aging on this relationship from lines 447-466.
In references to NKp44 and telomere length, as suggested we went over Delgado et al. J Med Genet 2018 (PMID 29151059) paper, the SNP (rs9357354) in the locus 6p21.1 is closest to the gene NCR2 which has not been previously linked to telomere length and their initial hit with this and the two other SNPs in the South Asian population was not replicated in the ENGAGE consortium cohort of European descent despite the populations having similar mean allele frequencies for the SNPs in questions. There is no published data showing an age-related change to NKp44 (the gene product of NCR2), thus, we were not able to confidently write about/address this comment.

However, we did find the topic of telomere length and telomerase activity in NK cells with age to be a very interesting perspective. We added a section on telomere length as a biomarker for aged and differentiated NK cells to lines 342-353 in the NK Cell Exhaustion section.

Round 2

Reviewer 1 Report

The authors addressed most of my comments and suggestions by editing the manuscript accordingly. I believe the article is acceptable for publication.